# MTGER: Multi-view Temporal Graph Enhanced Temporal Reasoning over Time-Involved Document

**Zheng Chu[1], Zekun Wang[1], Jiafeng Liang[1], Ming Liu[1,2*], Bing Qin[1,2]**
[1]Harbin Institute of Technology, Harbin, China
[2]Peng Cheng Laboratory, Shenzhen, China
{zchu, zkwang, jfliang, mliu,* qinb}@ir.hit.edu.cn

## Abstract

The facts and time in the document are intricately intertwined, making temporal reasoning over documents challenging. Previous work models time implicitly, making it difficult to handle such complex relationships. To address this issue, we propose **MTGER**, a novel **M**ulti-view **T**emporal **G**raph **E**nhanced **R**easoning framework for temporal reasoning over time-involved documents. Concretely, MTGER explicitly models the temporal relationships among facts by multi-view temporal graphs. On the one hand, the heterogeneous temporal graphs explicitly model the temporal and discourse relationships among facts; on the other hand, the multi-view mechanism captures both time-focused and fact-focused information, allowing the two views to complement each other through adaptive fusion. To further improve the implicit reasoning capability of the model, we design a self-supervised time-comparing objective. Extensive experimental results demonstrate the effectiveness of our method on the TimeQA and SituatedQA datasets. Furthermore, MTGER gives more consistent answers under question perturbations.

## 1 Introduction

In the real world, many facts change over time, and these changes are archived in the document such as Wikipedia. Facts and time are intwined in documents with complex relationships. Thus temporal reasoning is required to find facts that occurred at a specific time. To investigate this problem, Chen et al. (2021) propose the TimeQA dataset and Zhang and Choi (2021) propose the SituatedQA dataset. For example, Figure 1 illustrates a question involving implicit temporal reasoning. From the human perspective, to answer this question, we first need to find relevant facts in the document and obtain new facts based on existing facts (left of Figure 1(d)). We need to deduce the answer from

*Corresponding Author.

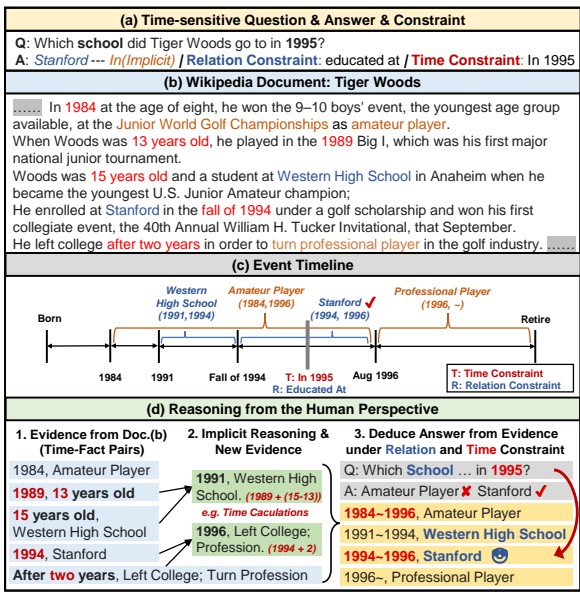

Figure 1: An example of a time-sensitive question involving implicit reasoning, best viewed in color. (a) describes the question, answer and constraints (b) shows a time-involved document (c) depicts the timeline of events in the document (d) illustrates the reasoning process in the human perspective, including three steps.

the existing facts according to the constraints of the question, including time constraints and relation constraints (right of Figure 1(d)).

Recent work (Chen et al., 2021; Zhang and Choi, 2021; Izacard and Grave, 2021b; Zaheer et al., 2020; Beltagy et al., 2020a; Guo et al., 2022) directly uses pre-trained models or large-scale language models to answer time-sensitive questions, neglecting to explicitly model the time like human, resulting in lack of understanding of time. Even the state-of-the-art QA models have a large gap compared with human performance (50.61 EM vs. 87.7 EM), indicating that there is still a long way to go in this research area.

Inspired by the human reasoning process depicted in Figure 1(d), we intend to explicitly model the temporal relationships among facts. To this

end, we propose Multi-view Temporal Graph Enhanced Reasoning framework (MTGER) for temporal reasoning over time-involved documents. We construct a multi-view temporal graph to establish correspondence between facts and time and to explicitly model the temporal relationships between facts. The explicit and implicit temporal relations between facts in the heterogeneous graph enhance the temporal reasoning capability, and the cross-paragraph interactions between facts alleviate the inadequate interaction.

Specifically, each heterogeneous temporal graph (HTG) contains factual and temporal layers. Nodes in the factual layer are events, and nodes in the temporal layer are the timestamps (or time intervals) corresponding to the events. Different nodes are connected according to the discourse relations and relative temporal relations. In addition, we construct a time-focused HTG and a fact-focused HTG to capture information with different focuses, forming a multi-view temporal graph. We complement the two views through adaptive fusion to obtain more adequate information. At the decoder side, we use a question-guided fusion mechanism to dynamically select the temporal graph information that is more relevant to the question. Finally, we feed the time-enhanced representation into the decoder to get the answer. Furthermore, we introduce a self-supervised time-comparing objective to enhance the temporal reasoning capability.

Extensive experimental results demonstrate the effectiveness of our proposed method on the TimeQA and SituatedQA datasets, with a performance boost of up to 9% compared to the state-of-the-art QA model and giving more consistent answers when encountering input perturbations.

The main contributions of our work can be summarized as follows:

- We propose to enhance temporal reasoning by modeling time explicitly. As far as we know, it is the first attempt to model time explicitly in temporal reasoning over documents.

- We devise a document-level temporal reasoning framework, MTGER, which models the temporal relationships between facts through heterogeneous temporal graphs with a complementary multi-view fusion mechanism.

- Extensive experimental results demonstrate the effectiveness and robustness of our method on the TimeQA and SituatedQA datasets.

## 2 MTGER Framework

### 2.1 Task Definition

Document-level textual temporal reasoning tasks take a long document (e.g., typically thousands of characters) with a time-sensitive question as input and output the answer based on the document. Formally, given a time-sensitive question $Q$, a document $D$, the goal is to obtain the answer $A$ which satisfies the time and relation constraints in question $Q$. The document $D = \{P_1, P_2, ..., P_k\}$ contains $k$ paragraphs, which are either from a Wikipedia page or retrieved from Wikipedia dumps. The answer can be either an extracted span or generated text, and we take the generation approach in this paper. Please refer to Appendix A.1 for the definition of time-sensitive questions.

$$A^* = \arg \max P(A \mid Q, D; \theta) \qquad (1)$$

### 2.2 Overview

As depicted in Figure 2, MTGER first encodes paragraphs and constructs a multi-view temporal graph, then applies temporal graph reasoning over the multi-view temporal graph with the time-comparing objective and adaptive fusion, and finally feeds the time-enhanced features into the decoder to get the answer.

### 2.3 Text Encoding and Graph Construction

**Textual Encoder** We use the pre-trained FiD (Izacard and Grave, 2021b) model to encode long text. FiD consists of a bi-directional encoder and a decoder. It encodes paragraphs individually at the encoder side and concatenates the paragraph representations as the decoder input.

Given a document $D = \{P_1, P_2, ..., P_k\}$ containing $K$ paragraphs, each paragraph contains $M$ tokens: $P_i = \{x_1, x_2, ..., x_m\}$, $h$ represents the hidden dimension. Following the previous method (Izacard and Grave, 2021a), the question and paragraph are concat in a "question: title: paragraph:" fashion. The textual representation $\boldsymbol{H}_{text} \in \mathbb{R}^{k \times m \times h}$ is obtained by encoding all paragraphs individually using the FiD Encoder.

$$\boldsymbol{H}_{text} = \text{Enc}(P_1, P_2, ..., P_k) \qquad (2)$$

**Graph Construction** We construct a multi-view heterogeneous temporal graph based on the relationship between facts and time in the document, as illustrated in Figure 2(a). The multi-view temporal graph consists of two views, $\mathcal{G}_{fact} = (\mathcal{V}, \mathcal{E}_{fact})$

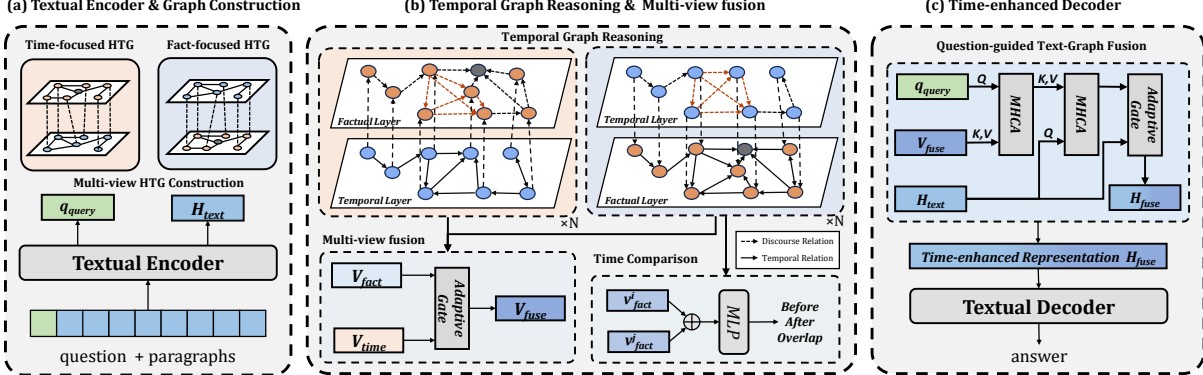

Figure 2: Overview of MTGER. Best viewed in color. (a) Encoding the context and constructing multi-view temporal graph (b) Temporal garph reasoning over multi-view temporal graph with time-comparing objective and adaptive multi-view fusion (c) Question-guided text-graph fusion and answer decoding.

and $\mathcal{G}_{time} = (\mathcal{V}, \mathcal{E}_{time})$, which has the same nodes and different edges.

**Procedure** Constructing a multi-view temporal graph consists of document segmentation, node extraction and edge construction. Firstly, we segment the document into chunks (or paragraphs) based on headings, with similar content within each chunk. Afterward, we extract time and fact nodes representing time intervals and events, using regular expressions. Finally, we construct edges based on the relationship between nodes and chunks. Besides, to mitigate graph sparsity problem, we introduce a global node to aggregate information among fact nodes. We will introduce nodes, edges, and views in the following.

**Nodes** According to the roles represented in temporal reasoning, we define six node types: **global node** is responsible for aggregating the overall information; **fact nodes** represent the events at the sentence-level granularity (e.g. He enrolled at Stanford); **time nodes** are divided into four categories according to their temporal states, before, after, between, and in, which represent the time interval of the events (e.g. Between 1980 and March 1988 [1980, 1998.25]). In the temporal graph, the global node and fact nodes are located at the factual layer and time nodes at the temporal layer.

**Edges** In the temporal layer, we build edges according to the temporal relationship between time nodes, including **before**, **after** and **overlap**. In the factual layer, we build edges according to the discourse relationship between fact nodes. For facts in the same paragraph, they usually share a common topic. Accordingly, we construct densely-connected intra-paragraph edges among these fact nodes. For facts in different paragraphs, we pick

two fact nodes in each of the two paragraphs to construct inter-paragraph edges. Temporal and factual layers are bridged by time-to-fact edges, which are uni-directed, from times to facts. The global node is connected with all fact nodes, from facts to the global node. The Appendix A.8 and A.9 provide an example of temporal graph and a more detailed graph construction process.

**Views** We construct two views, the **fact-focused** view and the **time-focused** view. Multiple views make it possible to model both absolute relationships between time expression (e.g. 1995 is before 2000 because 1995 < 2000) and relative relationships between events (e.g. Messi joins Inter Miami CF after his World Cup championship). If only one view exists, it is difficult to model both relationships simultaneously. In the time-focused view, time comparisons occur between time nodes, and fact nodes interact indirectly through time nodes as bridges; in the fact-focused view, the relative temporal relationships between facts are directly modeled. The model can sufficiently capture the temporal relationships among facts by complementing each other between the two views. To obtain the fact-focused view, we replace the discourse relation edges between fact nodes in the time-focused view with the temporal relation edges between the corresponding time nodes and replace the edges of time nodes with discourse relations of fact nodes in a similar way. The comparison of the two views is shown at the top of Figure 2(b).

## 2.4 Multi-view Temporal Graph Reasoning

**Temporal Graph Reasoning** First, we initialize the nodes representations using the text representation and then perform a linear transformation to

the nodes according to their types.

$$\boldsymbol{v}_i = \text{Pooling}(\boldsymbol{h}_1, \boldsymbol{h}_2, ..., \boldsymbol{h}_l) \qquad (3)$$

$$\boldsymbol{v}_i^{(0)} = \boldsymbol{W}^t \boldsymbol{v}_i \qquad (4)$$

$$\boldsymbol{V}^{(0)} = [\boldsymbol{v}_1^{(0)}, \boldsymbol{v}_2^{(0)}, ..., \boldsymbol{v}_n^{(0)}] \qquad (5)$$

where $\boldsymbol{h}_1, ..., \boldsymbol{h}_l$ are textual representations corresponding to the node v, $\boldsymbol{W}^t \in \mathbb{R}^{n \times n}$ is the linear transformation corresponding to node type t, n is number of nodes, $\boldsymbol{V}^{(0)} \in \mathbb{R}^{n \times h}$ serves as the first layer input to the graph neural network.

We refer to the heterogeneous graph neural network (Schlichtkrull et al., 2018; Busbridge et al., 2019) to deal with different relations between nodes. In this paper, we use heterogeneous graph attention network.

We define the notation as follows: $\boldsymbol{W}^r \in \mathbb{R}^{h \times h}, \boldsymbol{W}_Q^r \in \mathbb{R}^{h \times h}, \boldsymbol{W}_K^r \in \mathbb{R}^{h \times h}$ represents the node transformation, query transformation and key transformation under relation $r$, respectively. They are all learnable parameters.

First, we perform a linear transformation on the nodes according to the relations they are located.

$$\boldsymbol{O}^r = \boldsymbol{V}\boldsymbol{W}^r \in \mathbb{R}^{n \times h} \qquad (6)$$

$$\boldsymbol{Q}^r = \boldsymbol{O}^r \boldsymbol{W}_Q^r \in \mathbb{R}^{n \times h} \qquad (7)$$

$$\boldsymbol{K}^r = \boldsymbol{O}^r \boldsymbol{W}_K^r \in \mathbb{R}^{n \times h} \qquad (8)$$

Afterwards, we calculate the attention scores[1] in order to aggregate the information.

$$\text{a}_{i,j}^r = \boldsymbol{q}_i^r \cdot \boldsymbol{k}_j^r \qquad (9)$$

$$\alpha_{i,j}^r = \frac{\exp(\text{a}_{i,j}^r)}{\sum_{k \in \mathcal{N}_r(i)} \exp(\text{a}_{i,k}^r)} \qquad (10)$$

Finally, we obtain the updated node representation by aggregating based on the attention score. Here we use the multi-head attention mechanism (Vaswani et al., 2017), where $K$ is the number of attention heads.

$$\boldsymbol{V}^{(\ell+1)} = \bigoplus_{k=1}^{K} \sigma \left( \sum_{r \in \mathcal{R}} \sum_{j \in \mathcal{N}_r(i)} \alpha_{i,j}^{r,k} \boldsymbol{o}_j^{r,k} \right) \qquad (11)$$

We obtain the final graph representation through $L$ layer heterogeneous graph neural network.

$$\boldsymbol{V}^{(\ell+1)} = \text{HeteroGNN}(\boldsymbol{V}^{(\ell)}) \qquad (12)$$

---

[1]Formula of multi-head attention is omitted for readability.

**Adaptive Multi-view Graph Fusion**   Through the graph reasoning module introduced above, we can obtain the fact-focused and time-focused graph representation, $\boldsymbol{V}_f \in \mathbb{R}^{n \times h}, \boldsymbol{V}_t \in \mathbb{R}^{n \times h}$, respectively. These two graph representations have their own focus, and to consider both perspectives concurrently, we adopt an adaptive fusion manner (Li et al., 2022; Zhang et al., 2023) to model the interaction between them, as illustrated in Figure 2(b).

$$\lambda = \text{Sigmoid}(\boldsymbol{W}_f \boldsymbol{V}_f + \boldsymbol{W}_t \boldsymbol{V}_t) \qquad (13)$$

$$\boldsymbol{V}_{fuse} = (1 - \lambda) \cdot \boldsymbol{V}_f + \lambda \cdot \boldsymbol{V}_t \qquad (14)$$

where $\boldsymbol{W}_f \in \mathbb{R}^{h \times h}$ and $\boldsymbol{W}_t \in \mathbb{R}^{h \times h}$ are learnable parameters.

**Self-supervised Time-comparing Objective**   To further improve the implicit reasoning capability of models over absolute time, we design a self-supervised time-comparing objective. First, we transform the extracted TimeX into a time interval represented by a floating-point number. After that, the fact nodes involving TimeX in the graph are combined two by two. Three labels are generated according to the relationship between the two time intervals: before, after, and overlap. Assuming that there are $N$ fact nodes in the graph, $N(N-1)/2$ self-supervised examples can be constructed, and we use cross-entropy to optimize them.

$$h_{i,j} = \text{MLP}([h_i; h_j]) \qquad (15)$$

$$\mathcal{L}_{tc} = -\sum_{i=1}^{N-1} \sum_{j=i+1}^{N} y_{i,j} \log(P(y_{i,j}'|h_{i,j})) \qquad (16)$$

where $h_i, h_j$ are the nodes representations and $y_{i,j}$ is the pseudo-label. $[; ]$ denotes vector concatenate.

## 2.5   Time-enhanced Decoder

**Question-guided Text-graph Fusion**   Now we have the text representation $\boldsymbol{H}_{text}$ and the multi-view graph representation $\boldsymbol{V}_{fuse}$. Next, we dynamically fuse the text representation and the graph representation guided by the question to get the time-enhanced representation $\boldsymbol{H}_{fuse}$, which will be fed into the decoder to generate the final answer, as illustrated in Figure 2(c).

$$\boldsymbol{V}_{fuse}' = \text{MHCA}(\boldsymbol{q}_{query}, \boldsymbol{V}_{fuse}, \boldsymbol{V}_{fuse}) \qquad (17)$$

$$\boldsymbol{H}_{text}' = \text{MHCA}(\boldsymbol{H}_{text}, \boldsymbol{V}_{fuse}', \boldsymbol{V}_{fuse}') \qquad (18)$$

$$\boldsymbol{H}_{fuse} = \text{AdapGate}(\boldsymbol{H}_{text}, \boldsymbol{H}_{text}') \qquad (19)$$

where $\boldsymbol{q}_{query}$ is query embedding, MHCA is multi-head cross-attention, and AdapGate is mentioned in the graph fusion section.

**Time-enhanced Decoding** Finally, the time-enhanced representation $\boldsymbol{H}_{fuse}$ is fed into the decoder to predict the answer, and we use the typical teacher forcing loss to optimize the model.

$$\mathcal{L}_{tf} = -\sum_{i=1}^{N} y_i \log P(y_i' \mid \boldsymbol{H}_{fuse}, y_{<i}) \quad (20)$$

## 2.6 Training Objective

The training process has two optimization objectives: teacher-forcing loss for generating answers by maximum likelihood estimation and time-comparing loss for reinforcing time-comparing ability in the graph module. We use a multi-task learning approach to optimize them.

$$\mathcal{L}_{total} = \mathcal{L}_{tf} + \lambda \cdot \mathcal{L}_{tc} \quad (21)$$

where $\lambda$ is a hyper-parameter.

## 3 Experiments

### 3.1 Experiments Setup

**Dataset** We conduct experiments on the TimeQA and SituatedQA (Chen et al., 2021; Zhang and Choi, 2021) datasets. TimeQA is a document-level temporal reasoning dataset to facilitate temporal reasoning over time-involved documents. It is crowd-sourced based on Wikidata and Wikipedia. SituatedQA is an open-domain QA dataset where each question has a specified temporal context, which is based on the NQ-Open dataset.

Following the original experimental setup, we use EM and F1 on the TimeQA and EM on the SituatedQA dataset as evaluation metrics. In addition, we calculate metrics for each question type. For more information about the datasets, please refer to Appendix A.1.

**Baseline** We choose four typical long-context QA models, FiD, BigBird, LED and LongT5 (Izac-ard and Grave, 2021b; Zaheer et al., 2020; Belt-agy et al., 2020a; Guo et al., 2022), an open-domain QA system DPR (Karpukhin et al., 2020) and a close-book QA model BART (Lewis et al., 2020a) as baselines. FiD, BigBird and LongT5 are inited with pre-trained checkpoint on NaturalQues-tions (Kwiatkowski et al., 2019), TriviaQA (Joshi et al., 2017) and NewsQA (Trischler et al., 2017). Besides, large-scale language models (LLMs) have demonstrated excellent performance on reasoning

and QA tasks. In order to explore their capabilities on textual temporal reasoning, we also include the state-of-the-art LLM, GPT3 and Chat-GPT[2] (Ouyang et al., 2022; OpenAI, 2022) for comparison.

For the large-scale language model baseline, we sample 10 answers per question, @mean and @max represent the average and best results, respectively. We use gpt-3.5-turbo@max in subsequent experiments unless otherwise stated.

**Implementation Details** We use Pytorch framework, Huggingface Transformers for pre-trained models and PyG for graph neural networks. We use the base size model with hidden dimension 768 for all experiments. For all linear transformations in graph neural networks, the dimension is $768 \times 768$. The learning rate schedule strategy is warm-up for the first 20% of steps, followed by cosine decay. We use AdamW as the optimizer (Loshchilov and Hutter, 2019). All experiments are conducted on a single Tesla A100 GPU. The training cost of TimeQA and SituatedQA is approximately 4 hours and 1.5 hours, respectively. Please refer to Appendix A.3 for detailed hyperparameters.

### 3.2 Main Result

**Results on TimeQA** Table 1 shows the overall experiment results of baselines and our methods on the TimeQA dataset. The first part shows the results of the LLM baselines. Even though the LLMs demonstrate their amazing reasoning ability, it performs poorly in temporal reasoning. The second part shows the results of supervised baselines, where BigBird performs on par with gpt-3.5-turbo, and FiD performs better than BigBird.

Our method MTGER outperforms all baselines, achieving 60.40 EM / 69.44 F1 on the easy split and 53.19 EM / 61.42 F1 on the hard split. MT-GER obtains 2.43 EM / 1.92 F1 and 3.39 EM / 2.95 F1 performance boost compared to the state-of-the-art QA model, FiD. We think this significant performance boost comes from explicit modelling of time and interaction across paragraphs. In addition, MTGER++ further improves the performance through long-context adaptation pre-training for the reader model, achieving 60.95 EM / 69.89 F1 on the easy split and 54.11 EM / 62.40 F1 on the hard split. Please refer to Appendix A.4 for more details about MTGER++.

---

[2]ChatGPT May 24 Version

| Method | Easy | | | | Hard | | | |
|---|---|---|---|---|---|---|---|---|
| | **Dev** | | **Test** | | **Dev** | | **Test** | |
| | **EM** | **F1** | **EM** | **F1** | **EM** | **F1** | **EM** | **F1** |
| **LLM Baseline** | | | | | | | | |
| **Zero-shot** | | | | | | | | |
| gpt-3.5-turbo@mean | 30.71 | 42.67 | 32.79 | 46.04 | 28.91 | 36.94 | 28.68 | 35.14 |
| gpt-3.5-turbo@max | 43.14 | 53.92 | 43.46 | 56.18 | 37.14 | 45.08 | 37.64 | 43.32 |
| **Few-shot(n=5)** | | | | | | | | |
| gpt-3.5-turbo@mean | 42.14 | 51.59 | 34.80 | 47.12 | 35.37 | 43.52 | 34.03 | 42.36 |
| gpt-3.5-turbo@max | 54.67 | 62.92 | 44.98 | 58.66 | 45.69 | 54.41 | 41.63 | 51.71 |
| **Supervised Baseline** | | | | | | | | |
| LED | 46.97 | 56.41 | 49.41 | 58.35 | 39.78 | 48.45 | 40.50 | 48.10 |
| LongT5 | 48.17 | 59.59 | 48.14 | 58.94 | 41.43 | 52.58 | 42.94 | 53.50 |
| BigBird$_{tqa}$ | 52.00 | 61.13 | 50.65 | 60.21 | 43.69 | 51.71 | 44.73 | 51.59 |
| BigBird$_{nq}$ | 51.89 | 62.43 | 50.61 | 60.88 | 44.04 | 53.83 | 43.56 | 53.60 |
| FiD$_{tqa}$ | 55.21 | 65.82 | 56.32 | 64.99 | 47.62 | 56.74 | 47.92 | 56.78 |
| FiD$_{nq}$ | 57.25 | 66.56 | 58.06 | 67.52 | 49.21 | 58.00 | 49.80 | 58.74 |
| **Ours** | | | | | | | | |
| MTGER | **59.65** | **68.05** | 60.49 | 69.44 | **52.31** | 60.66 | 53.19 | 61.42 |
| MTGER++ | 59.20 | 67.70 | **60.95** | **69.89** | 52.02 | **61.06** | **54.15** | **62.40** |

Table 1: Main results on the TimeQA Dataset. All supervised models are base size and we report the average results of three runs. Best and second results are highlighted by **bold** and underline.

| Method | Easy | | Hard | | Perf. Gap | |
|---|---|---|---|---|---|---|
| | **EM** | **F1** | **EM** | **F1** | **EM** | **F1** |
| **LLM** | | | | | | |
| Zeroshot | 43.45 | 56.94 | 36.28 | 43.06 | -1.73% | +0.5% |
| Fewshot(n=5) | 52.74 | 64.75 | 50.63 | 57.80 | +19.4% | +11.1% |
| **Supervised** | | | | | | |
| BigBird$_{nq}$ | 46.81 | 55.67 | 41.75 | 51.36 | -5.8% | -6.4% |
| FiD$_{nq}$ | 55.91 | 63.86 | 47.52 | 54.82 | -4.3% | -6.4% |
| **Ours** | | | | | | |
| MTGER | 58.74 | 66.12 | 50.55 | 57.05 | **-3.9%** | **-5.9%** |
| MTGER++ | **59.35** | **66.51** | **51.06** | **57.91** | -4.1% | -6.0% |

Table 2: Results on TimeQA Human-Paraphrased dataset. Gap represents the performance gap compared with the main dataset.

| Method | Temp | | | |
|---|---|---|---|---|
| | **Static** | **Samp.** | **Start** | **Avg.** |
| **LLM** | | | | |
| Zeroshot | 11.6 | 8.5 | 11.9 | 9.9 |
| Fewshot(n=5) | 9.3 | 13.4 | 15.5 | 13.4 |
| **Supervised Baseline** | | | | |
| LED | 18.1 | 16.2 | 20.4 | 17.8 |
| LongT5 | 22.0 | 14.4 | 18.3 | 16.8 |
| BART | 26.0 | 16.2 | 18.3 | 18.3 |
| DPR | 39.8 | 17.2 | 24.9 | 23.0 |
| BigBird$_{nq}$ | 34.2 | 15.7 | 21.3 | 20.2 |
| FiD$_{nq}$ | 28.7 | 22.2 | **31.8** | 26.4 |
| **Ours** | | | | |
| MTGER | **45.0** | **24.8** | 28.2 | **28.8** |

Table 3: Results on the SituatedQA Temp test set. Evaluation metric is EM from the correct temporal context.

**Results on TimeQA Human-Paraphrased** The questions in TimeQA human-paraphrased are rewritten by human workers in a more natural language manner, making the questions more diverse and fluent. As shown in Table 2, the performance of the LLMs increases rather than decreases, which we think may be related to the question becoming more natural and fluent. All supervised baselines show some performance degradation. Our method still outperforms all baselines and has less performance degradation compared with supervised baselines. The experimental results illustrate that our method is able to adapt to more diverse questions.

**Results on SituatedQA** To investigate the generalization of MTGER for temporal reasoning, we conduct experiments on an open-domain temporal reasoning dataset, SituatedQA. Since the model initiated with NaturalQuestion checkpoint performs better than TriviaQA, we do not present the results for models initiated with TriviaQA.

As shown in Table 3, MTGER achieves 28.8 EM, exceeding all baseline methods, improving by 9% compared to the best baseline. MTGER does not have a reranking module while it outperforms DPR by 5.8 EM, which has a reranking module. Our method performs well on all three types of questions and significantly better than others on Static and Samp. types. The experimental results demonstrate that our method can be generalized to

| Method | Hard-Implicit | | | | | Easy-Explicit | | |
|--------|------|---------|--------|-------|------|------|---------|------|
| | **in** | **between** | **before** | **after** | **avg.** | **in** | **between** | **avg.** |
| **LLM Baseline** | | | | | | | | |
| Zeroshot | 38.79 | 34.53 | 41.46 | 27.50 | 36.69 | 33.33 | 44.39 | 43.46 |
| Fewshot(n=5) | 37.69 | 45.36 | 56.09 | 25.50 | 41.41 | 45.27 | 40.47 | 44.86 |
| **Supervised Baseline** | | | | | | | | |
| BigBird$_{nq}$ | 46.31 | 45.04 | 40.22 | 42.36 | 44.85 | 28.97 | 52.34 | 50.61 |
| FiD$_{nq}$ | 49.61 | 48.83 | 50.18 | 53.81 | 49.73 | 56.10 | 58.22 | 58.06 |
| **Ours** | | | | | | | | |
| MTGER | **52.99** | 52.80 | 53.87 | 56.10 | 53.28 | **57.92** | 60.69 | 60.49 |
| MTGER++ | **52.99** | **53.57** | **56.83** | **57.25** | **53.99** | 57.47 | **61.23** | **60.95** |

Table 4: Results for different question types on the TimeQA test set. Evaluation metric is EM.

different temporal reasoning datasets and achieves excellent performance.

## 4 Analysis

### 4.1 Probe Different Question Types

We calculate the metrics for different categories of implicit questions in the hard split and explicit questions in the easy split. Results are shown in Table 4. Our method outperforms all baselines on all question types, achieving 53.99 Implicit-EM and 60.95 Explicit-EM, compared to the best baseline of 49.73 Implicit-EM and 58.06 Explicit-EM, reaching a relative improvement of 8.6% for implicit questions and 5% for explicit questions.

According to the results, our method is particularly good at handling hard questions that require implicit reasoning. We believe the explicit temporal modelling provided by the heterogeneous temporal graphs improves the implicit temporal reasoning capability of models. In the next section, we will perform an ablation study on different modules of MTGER to verify our conjecture.

### 4.2 Ablation Study

We investigate the effects of text-graph fusion, time-comparing objective and multi-view temporal graph. For multi-view temporal graphs, we conduct a more detailed analysis by sequentially removing the multi-view graph, heterogeneous temporal graph and homogeneous temporal graph. Experimental results are shown in Table 5.

**Effect of Text-graph Fusion** Question-guided text-graph fusion module dynamically selects the graph information that is more relevant to the question. As shown in Table 5(a), removing this module results in the performance degradation of 0.54 EM / 0.43 EM and 0.87 EM / 0.80 EM in the dev set

| Method | Dev | | Test | |
|--------|------|------|------|------|
| | **Easy** | **Hard** | **Easy** | **Hard** |
| MTGER | **59.65** | **52.31** | 60.49 | **53.19** |
| - Text-graph Fusion (a) | 59.11 | 51.88 | 59.52 | 52.39 |
| - Time-comparing (b) | **59.65** | 51.53 | **60.66** | 52.65 |
| - Multi-view Graph (c.1) | 59.17 | 52.04 | 60.12 | 52.42 |
| - HeteroGraph (c.2) | 58.67 | 50.52 | 59.89 | 50.78 |
| - HomoGraph (c.3) | 57.25 | 49.21 | 58.06 | 49.80 |

Table 5: Results of Ablation Study on the TimeQA dataset. Evaluation metric is EM.

and test set, respectively. This suggests that the question-guided text-fusion module can improve overall performance by dynamically selecting more useful information.

**Effect of Time-comparing Objective** As shown in Table 5(b), removing the time-comparing objective has almost no effect on the easy split, and the performance degradation in the dev and test set on the hard split is 0.78 EM / 0.54 EM, indicating that the time-comparing objective mainly improves the implicit temporal reasoning capability of models.

**Effect of Multi-view Temporal Graph** We sequentially remove the multi-view temporal graph (c.1 keep only one heterogeneous temporal graph); replace the heterogeneous temporal graph with a homogeneous temporal graph (c.2 do not distinguish the relations between nodes); and remove the graph structure (c.3) to explore the effect of graph structure in detail.

Removing the multi-view temporal graph (c.1) brings an overall performance degradation of 0.48 EM / 0.78 EM and 0.37 EM / 0.77 EM in the dev set and test set, respectively, implying that the complementary nature of multi-view mechanism helps to capture sufficient temporal relationships between facts, especially implicit relationships.

| Perturbated Question | Context | FiD | MtGer |
|---|---|---|---|
| **Q**: Which school did Beatrix Tugendhut Gardner go to in Dec 1955? **Q′**: Which school did Beatrix Tugendhut Gardner go to before 1954 and 1955? **Answer**: Brown University | ... Beatrix , often spelled Beatrice , attended Radcliffe College in Massachusetts and received her bachelors degree in 1954 . In 1956 , she earned her masters degree from Brown University , working with Carl Pfaffman . She completed her PhD in zoology at Oxford University in 1959 where she studied under the mentorship of Niko Tinbergen ... | **Answer′**: Radcliffe College ✗ | **Answer′**: Brown University ✔ |
| **Q**: Which school did Jay Rockefeller go to between Oct 1966 and Dec 1967? **Q′**: Which school did Jay Rockefeller go to before 1973? **Answer**: Unanswerable | ... Rockefeller moved to Emmons , West Virginia , to serve as a VISTA worker in 1964 and was first elected to public office as a member of the West Virginia House of Delegates ( 1966-1968 ) . Rockefeller was later elected West Virginia Secretary of State ( 1968-1973 ) and was president of West Virginia Wesleyan College ( 1973–1975 ) . He became the states senior U.S . Senator when the long-serving Senator Robert Byrd died in June 2010 . ... | **Answer′**: West Virginia Wesleyan College ✗ | **Answer′**: Unanswerable ✔ |

Table 6: Case study from the consistency analysis. **Q** stands for the original question, **Q′** stands for the perturbated question and **Answer′** stands for the answer after question perturbation.

We replace the heterogeneous temporal graph with a homogeneous temporal graph (c.2), which results in the GNN losing the ability to explicitly model the temporal relationships between facts, leaving only the ability to interact across paragraphs. The performance degradation is slight in the easy split, while it causes significant performance degradation of 1.52 EM / 1.64 EM compared with (c.1) in the hard split, which indicates that explicit modelling of temporal relationships between facts can significantly improve the implicit reasoning capability.

Removing the graph structure also means removing both text-graph fusion and time-comparing objective, which degrades the model to a FiD model (c.3). At this point, the model loses the cross-paragraph interaction ability, and there is an overall degradation in performance, which suggests that the cross-paragraph interaction can improve the overall performance by establishing connections between facts and times.

### 4.3 Consistency Analysis

To investigate whether the model can consistently give the correct answer when the time specifier of questions is perturbed, we conduct a consistency analysis. The experimental results in Table 7 show that our method exceeds the baseline by up to 18%, which indicates that our method is more consistent and robust compared to baselines. Please refer to Appendix A.7 for details of consistency analysis.

| Method | Consistency |
|---|---|
| BigBird$_{nq}$ | 0.63 |
| FiD$_{nq}$ | 0.66 |
| text-davinci-003@mean | 0.54 |
| gpt-3.5-turbo@mean | 0.46 |
| MtGer | 0.76 |
| MtGer++ | **0.78** |

Table 7: Results of consistency analysis on the TimeQA hard test set. The higher the consistency, the better.

### 4.4 Case Study

We show two examples from the consistency analysis to illustrate the consistency of our model in the face of question perturbations, as shown in Table 6. Both examples are from the TimeQA hard dev set.

The first example shows the importance of implicit temporal reasoning. From the context, we know that *Beatrix* received her bachelor's degree in 1954 and her master's degree in 1956. The master's came after the bachelor's, so we can infer that she was enrolled in a master's degree between 1954 and 1956, and 1954-1955 lies within this time interval, so she was enrolled in *Brown University* during this period. Since FiD lacks explicit temporal modelling, its implicit temporal reasoning ability is weak and fails to predict the correct answer.

The second example shows the importance of question understanding. The question is about which school he attends, which is not mentioned in the context. FiD incorrectly interprets the question as to where he is working for, fails to understand the question and gives the wrong answer. Our method,

which uses a question-guided fusion mechanism, allows for a better question understanding and consistently gives the correct answer.

# 5 Related Work

## 5.1 Temporal Reasoning in NLP

**Knowledge Base Temporal Reasoning** Knowledge base QA retrieves facts from a knowledge base using natural language queries (Berant et al., 2013; Bao et al., 2016; Lan and Jiang, 2020; He et al., 2021; Srivastava et al., 2021). In recent years, some benchmarks specifically focus on temporal intents, including TempQuestions (Jia et al., 2018a) and TimeQuestions (Jia et al., 2021). TEQUILA (Jia et al., 2018b) decomposes complex questions into simple ones by heuristic rules and then solves simple questions via general KBQA systems. EXAQT (Jia et al., 2021) uses Group Steiner Trees to find subgraph and reasons over subgraph by RGCN. SF-TQA (Ding et al., 2022) generates query graphs by exploring the relevant facts of entities to retrieve answers.

**Event Temporal Reasoning** Event Temporal Reasoning focuses on the relative temporal relationship between events, including event temporal QA (Ning et al., 2020; Lu et al., 2022; Han et al., 2021), temporal commonsense reasoning (Qin et al., 2021; Zhou et al., 2019, 2020), event timeline extraction (Faghihi et al., 2022), and temporal dependency parsing (Mathur et al., 2022). TranCLR (Lu et al., 2022) injects event semantic knowledge into QA pipelines through contrastive learning. ECONET (Han et al., 2021) equips PLMs with event temporal relations knowledge by continuing pre-training. TACOLM (Zhou et al., 2020) exploits explicit and implicit mentions of temporal commonsense by sequence modelling.

**Textual Temporal Reasoning** Textual temporal reasoning pays more attention to the temporal understanding of the real-world text, including both absolute timestamps (e.g. before 2019, in the late 1990s) and relative temporal relationships (e.g. A occurs before B). To address this challenge, Chen et al. (2021) proposes the TimeQA dataset, and Zhang and Choi (2021) propose the SituatedQA dataset. Previous work directly adopts long-context QA models (Izacard and Grave, 2021b; Zaheer et al., 2020) and lacks explicit temporal modelling.

In this paper, we focus on temporal reasoning over documents and explicitly model the temporal

relationships between facts by graph reasoning over multi-view temporal graph.

## 5.2 Question Answering for Long Context

Since the computation and memory overhead of Transformer-based models grows quadratically with the input length, additional means are required to reduce the overhead when dealing with long-context input. ORQA (Lee et al., 2019) and RAG (Lewis et al., 2020b) select a small number of relevant contexts to feed into the reader through the retriever, but much useful information may be lost in this way. FiD (Izacard and Grave, 2021b) and M3 (Wen et al., 2022) reduce the overhead at the encoder side by splitting the context into paragraphs and encoding them independently. However, there may be a problem of insufficient interaction at the encoder side, and FiE (Kedia et al., 2022) introduces a global attention mechanism to alleviate this problem. Longformer, LED, LongT5, and BigBird (Beltagy et al., 2020b; Guo et al., 2022; Zaheer et al., 2020) reduce the overhead by sliding window and sparse attention mechanism.

# 6 Conclusion

In this paper, we devise MTGER, a novel temporal reasoning framework over documents. MTGER explicitly models temporal relationships through multi-view temporal graphs. The heterogeneous temporal graphs model the temporal and discourse relationships among facts, and the multi-view mechanism performs information integration from the time-focused and fact-focused perspectives. Furthermore, we design a self-supervised objective to enhance implicit reasoning and dynamically aggregate text and graph information through the question-guided fusion mechanism. Extensive experimental results demonstrate that MTGER achieves better performance than state-of-the-art methods and gives more consistent answers in the face of question perturbations on two document-level temporal reasoning benchmarks.

# Limitations

Although our proposed method exhibits excellent performance in document-level temporal reasoning, the research in this field still has a long way to go. We will discuss the limitations as well as possible directions for future work. First, the automatically constructed sentence-level temporal graphs are slightly coarse in granularity; a fine-grained

temporal graph can be constructed by combining an event extraction system in future work to capture fine-grained event-level temporal clues accurately. Second, our method does not give a temporal reasoning process, and in future work, one can consider adding a neural symbolic reasoning module to provide better interpretability.

## Acknowledgements

The research in this article is supported by the National Key Research and Development Project (2022YFF0903301), the National Science Foundation of China (U22B2059, 61976073, 62276083), and Shenzhen Foundational Research Funding (JCYJ20200109113441941), Major Key Project of PCL (PCL2021A06).

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

# A Appendix

## A.1 Datasets Details

**TimeQA**  TimeQA contains a main dataset and a human-paraphrased dataset, where the questions in the main dataset are synthesized by the templates the questions in the human-paraphrased dataset are rewritten manually by humans. The questions in the dataset include both explicit and implicit forms. Explicit question types contain **in-explicit** and **between-explicit**, and implicit question types contain **in-implicit, between-implicit, before-implicit, and after-implicit**. The easy split contains only explicit questions, and the hard split contains most implicit questions and a few explicit questions. The statistics of the dataset are shown in Table 8 and Table 9.

**SituatedQA**  The same question may have different answers depending on the context. SituatedQA is an open-domain QA dataset that requires a specific context to produce the correct answer. The dataset contains two types of context, temporal and geographic, and we use the questions with the temporal context in this paper. The evaluation metrics in the original paper include EM-One (the answer exactly matches the correct context) and EM-Any (the answer matches any of the annotated contexts), and we use EM-One as the evaluation metric. The original task form of SituatedQA is open-domain QA, and we transform it into a long-context QA by retrieving relevant fragments using a dense retriever. Please refer to Appendix A.2 for the retrieval details.

**Explicit and Implicit Questions**  Explicit Questions: The timestamps in the questions appear in the document. Implicit Questions: (a) the timestamps in the questions do not appear in the document (b) vague timestamps, such as the 1990s and 21st century (c) the timestamps involving commonsense knowledge, such as World War II lasted from 1939 to 1945.

**Time-sensitive Questions**  We refer to the definition in (Chen et al., 2021): (a) each question contains a time identifier (b) changing the time identifier causes the answer to change (c) it requires temporal reasoning to answer the question.

## A.2 Details of SituatedQA Retrieval

Following the DPR (Karpukhin et al., 2020), we divide Wikipedia into segments by no overlapping

| Dataset | #Train | #Dev | #Test |
|---|---|---|---|
| TimeQA Easy | 14308 | 3021 | 2997 |
| TimeQA Hard | 14681 | 3087 | 3078 |
| TimeQA Human-Paraphrased Easy | 1171 | - | 989 |
| TimeQA Human-Paraphrased Hard | 1171 | - | 989 |
| SituatedQA Temp | 6009 | 3423 | 2795 |

Table 8: Statistic of TimeQA and SituatedQA.

| Split | Implicit | | | | Explicit | |
|---|---|---|---|---|---|---|
| | in | bet. | bef. | aft. | bet. | in |
| Easy | - | - | - | - | 92.43% | 7.56% |
| Hard | 37.75% | 38.79% | 8.32% | 7.75% | 7.37% | - |

Table 9: The proportion of the #question of different categories in the TimeQA training set.

windows of length 100. We take the Pyserini (Lin et al., 2021) library, using DPR as the dense retriever[3]. We select the top 20 most similar segments for each question as the context. The English Wikipedia dumps we used is as of Feb 20, 2021.

## A.3 Hyperparameters

The hyperparameters we used during our experiments are shown in Table 10. We search through the listed hyperparameters and end up using the bolded ones in each row.

| Hyperparameters | Values |
|---|---|
| Learning Rate | 2e-5 **5e-5** |
| Batch Size | **4** |
| Warmup | **20%** |
| Num GNN Layers | 3 **5** 7 |
| Num Graph Relations | 4 **6** 8 |
| GNN Dropout | 0.1 **0.5** 0.9 |
| Beam Size | **1** |
| Training Epochs | **3** |
| $\lambda$ | 0.01 **0.001** |

Table 10: Hyperparameters

## A.4 Stronger Reader with Long-context Adaptation

We perform long-context adaptation pre-training for models that do not support long-context input. We replace the reader backbone with UnifiedQA (Khashabi et al., 2020), a strong QA model. However, UnifiedQA is built on vanilla T5 (Raffel et al., 2020) and does not support long context input, which needs additional adaptations. We use a similar approach to FiD (Izacard and Grave,

---

[3]https://huggingface.co/facebook/dpr-ctx_encoder-multiset-base

| Method | Easy | | | | Hard | | | |
|---|---|---|---|---|---|---|---|---|
| | Dev | | Test | | Dev | | Test | |
| | EM | F1 | EM | F1 | EM | F1 | EM | F1 |
| **Zero-shot** | | | | | | | | |
| gpt-3.5-turbo@mean | 30.71 | 42.67 | 32.79 | 46.04 | 28.91 | 36.94 | 28.68 | 35.14 |
| gpt-3.5-turbo@max | **43.14** | **53.92** | **43.46** | **56.18** | **37.14** | **45.08** | 37.64 | 43.32 |
| text-davinci-003@mean | 21.27 | 34.59 | 24.34 | 33.47 | 22.21 | 27.27 | 23.80 | 30.52 |
| text-davinci-003@max | 32.40 | 49.05 | 38.02 | 51.09 | 34.26 | 43.20 | **38.44** | **46.83** |
| **Few-shot(n=5)** | | | | | | | | |
| gpt-3.5-turbo@mean | 42.14 | 51.59 | 34.80 | 47.12 | 35.37 | 43.52 | 34.03 | 42.36 |
| gpt-3.5-turbo@max | **54.67** | **62.92** | **44.98** | **58.66** | **45.69** | **54.41** | 41.63 | **51.71** |
| text-davinci-003@mean | 32.00 | 43.58 | 28.87 | 38.28 | 30.96 | 36.82 | 29.05 | 33.78 |
| text-davinci-003@max | 42.54 | 53.91 | 40.64 | 51.70 | 39.87 | 48.43 | **42.23** | 48.58 |
| **Ours** | | | | | | | | |
| MTGER | **59.65** | **68.05** | 60.49 | 69.44 | **52.31** | 60.66 | 53.19 | 61.42 |
| MTGER++ | 59.20 | 67.70 | **60.95** | **69.89** | 52.02 | **61.06** | **54.15** | **62.40** |

Table 11: Entire results of LLM baseline on the TimeQA dataset.

2021b), forcing the model to encode paragraphs separately at the encoder side, delaying the interaction across paragraphs to the decoder side, and use a FiD-style input format for training 3000 steps on long-context QA task for adaption. Finally, we obtain a UnifiedQA model adapted to long-context input. We replace the reader of MTGER with the long-context adapted UnifiedQA to obtain MTGER++.

## A.5 Prompts

The prompts we use are shown in Figure 4, and Figure 5. We try three different prompts and choose the one that performs best on the TimeQA dev set. We maintain a few-shot examples pool. For the k-shot prompt, k examples are drawn from it at a time, and we ensure that these k examples can cover all question types in the current dataset split.

Due to the maximum length limitation of the LLMs, the few-shot examples cannot take the full context. Therefore, we filter out some context irrelevant to the answer based on the original annotation data to ensure that the context does not exceed the maximum length limit.

## A.6 Entire Results of LLMs

We conduct experiments using gpt-3.5-turbo[4] (OpenAI, 2022) and text-davinci-003 (Ouyang et al., 2022) on the TimeQA dataset. The entire experiment results are shown in Table 11. Gpt-3.5-turbo has significantly outperformed text-davinci-003 in most cases, but they still have a large gap compared to our method.

---

[4]We use gpt-3.5-turbo (May 24 Version). https://platform.openai.com/

## A.7 Consistency Analysis

This section describes how to construct perturbated questions. A question and a time interval $[s, e]$ can determine the answer. We randomly sample a time interval $[s', e'] \subset [s, e]$, which ensures that the answer to the question does not change. After that, we randomly select 100 questions that the model answers correctly (EM=1.0), perturb the timestamps of the questions, and calculate the results after the perturbation. If the model still gives the correct answer after the perturbation, it is consistent; otherwise, it is inconsistent.

$$\text{consistency} = \frac{\#\text{cons.}}{\#\text{cons.} + \#\text{incons.}} \quad (22)$$

## A.8 An Example of Temporal Graph

Figure 3 illustrates a heterogeneous temporal graph under the time-focused view, with the legend on the right. There are three kinds of edges in the factual layer: **Intra-para Fact Edge** connects fact nodes within the same paragraph, **Inter-para Fact Edge** connects fact nodes across paragraphs and **Fact Aggregation Edge** connects fact nodes and the global node. There are three types of edges in the temporal layer, Before, After and Overlap, depending on the relative temporal relations between time nodes. Among them, Before and After are inverse edges of each other. **Time-to-Fact Edge** is the cross-layer edge for bridging fact and time nodes.

## A.9 Graph Construction Details

**Document Segmentation** We divide the document into chunks (or paragraphs) based on chapter

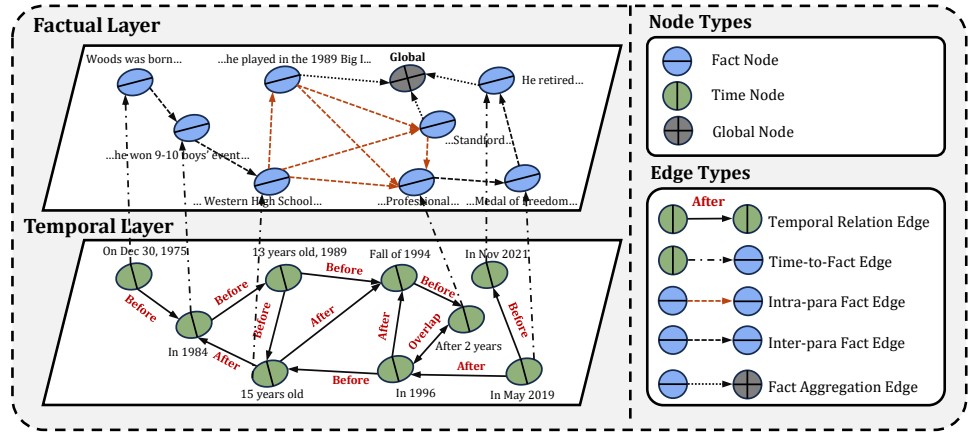

Figure 3: An example of heterogeneous temporal graph under time-focused view. Best viewed in color.

headings, with similar content within each paragraph. (e.g. Early life, College career, Profession career, Retirement, etc.)

**Node Extraction**  We design regular expressions to extract Time Expression from documents. We treat the extracted TimeX as time nodes containing 4 categories (In, Between, Before, After). The sentence which time nodes are located are treated as fact nodes, corresponding to the time nodes.

Examples: time nodes (In 1999, Before March 1988, etc.), fact nodes (He went to UCLA, etc.)

**Edge Construction**  Take a time-focused graph as an example. The graph contains a temporal layer and a factual layer.

**Factual Layer:** We construct fact edges according to the document paragraphs divided in the first step. Dense connections (Inter-para Fact Edge) are taken for fact nodes within the same paragraph (two nodes in the same paragraph), and sparse connections (Intra-para Fact Edge) are taken between fact nodes across paragraphs (two nodes are located in two adjacent paragraphs).

**Temporal Layer:** Each time node can be represented by a time interval (e.g. between 1923 and 1924 can be represented as [1923, 1924]; before March 1988 can be represented as $[-\infty, 1998.25]$). There are three kinds of interval relations between two time nodes: before, after and overlap. We use this to connect time nodes (Temporal Relation Edge). It is worth noting that the connections between time nodes are also dense within paragraphs and sparse across paragraphs.

**Other connections:** To mitigate the graph sparsity problem, we introduce a global node to aggregate information, and global nodes are connected to all fact nodes (Fact Aggr. Edge). Unidirectional connections are taken from fact to time nodes (Time-to-Fact Edge).

```
┌────────────────────────────────────────────────────────────────────────────────┐
│                               Zero-shot Prompt                                   │
├────────────────────────────────────────────────────────────────────────────────┤
│                                                                                  │
│  [User]                                                                          │
│  I will give you a question with context.                                        │
│  You need to answer my question based on the context.                            │
│  If you can infer the answer from the context, then output your answer; otherwise, if there is no answer, output │
│  [unanswerable].                                                                  │
│  Do not output anything else.                                                    │
│  question:                                                                       │
│  context:                                                                        │
│                                                                                  │
│  [Assistant]                                                                     │
│  answer: Model generated...                                                      │
│                                                                                  │
└────────────────────────────────────────────────────────────────────────────────┘
```

Figure 4: An example of zero-shot prompt.

```
┌────────────────────────────────────────────────────────────────────────────────┐
│                                Few-shot Prompt                                   │
├────────────────────────────────────────────────────────────────────────────────┤
│                                                                                  │
│  [User]                                                                          │
│  I will give you a question with context.                                        │
│  You need to answer my question based on the context.                            │
│  If you can infer the answer from the context, then output your answer; otherwise, if there is no answer, output │
│  [unanswerable].                                                                  │
│  Do not output anything else.                                                    │
│  question: Where was Clarice Phelps educated from 2016 to 2019?                  │
│  context: Clarice Phelps Clarice Evone Phelps is an American nuclear chemist researching the processing of │
│  radioactive transuranic elements at the U.S. ......                             │
│  answer: [unanswerable]                                                          │
│                                  ......                                           │
│                             Several Examples                                     │
│                                  ......                                           │
│  question: Which team did the player Michael Rankine belong to before Feb 2002?  │
│  context: Michael Rankine Michael Lee Rankine ( born 15 January 1985 ) is an English former professional │
│  footballer who played as a striker . Early career . ......                      │
│  answer: Doncaster Rovers                                                        │
│                                                                                  │
│  question:                                                                       │
│  context:                                                                        │
│                                                                                  │
│  [Assistant]                                                                     │
│  answer: Model generated...                                                      │
│                                                                                  │
└────────────────────────────────────────────────────────────────────────────────┘
```

Figure 5: An example of few-shot prompt.