# OpenReview forum: "MTGER: Multi-view Temporal Graph Enhanced Temporal Reasoning over Time-Involved Document"
_EMNLP/2023/Conference — EMNLP 2023 Findings_

### Official Review · Reviewer_DJLB · 2023-07-27

**Soundness:** 3

**Excitement:**

3: Ambivalent: It has merits (e.g., it reports state-of-the-art results, the idea is nice), but there are key weaknesses (e.g., it describes incremental work), and it can significantly benefit from another round of revision. However, I won't object to accepting it if my co-reviewers champion it.

**Paper Topic And Main Contributions:**

This paper proposes a multi-view temporal graph enhanced reasoning framework for temporal reasoning over time-involved documents, which explicitly models the temporal relationships among facts.

**Reasons To Accept:**

The studied task, document-level textual temporal reasoning, is important. Extensive experiments demonstrate the effectiveness of the proposed method to a certain degree.

**Reasons To Reject:**

There are some main weaknesses:
1. The design of the proposed method is not well-motivated. Why to segment heterogeneous temporal graph into two views and then fuse them? Keeping all nodes and edges in a single graph is better for information integrity consideration.
2. State-of-the-art QA baselines after 2022 are needed. Besides, why to use different baselines in different QA experiments.
3. The experimental analyses are not sufficient. For example, in Table 3, the proposed method is poorer than FiD_{nq} on Start type and in Table 5, removing time comparing is better than applying it on the easy split of the test set. However, no further explanation is provided.

**Reproducibility:**

3: Could reproduce the results with some difficulty. The settings of parameters are underspecified or subjectively determined; the training/evaluation data are not widely available.

**Reviewer Confidence:**

4: Quite sure. I tried to check the important points carefully. It's unlikely, though conceivable, that I missed something that should affect my ratings.

---

> ### Author Rebuttal · Authors · 2023-08-27
>
> Thank you for the careful and valuable review, which plays an important role in improving our paper! We address and clarify the questions and concerns below.
>
> **Re: Why to segment heterogeneous temporal graph into two views and then fuse them?**
>
> 1. Motivation: Multiple views make it possible to model both absolute relationships between time expression (e.g. 1995 is before 2000 because 1995 < 2000) and relative relationships between events (e.g. Messi joins Inter Miami CF after his World Cup championship). If there is only one view, it is difficult to model both relationships simultaneously.
> 2. Explanation:
>    1. Modeling relationships between time nodes (time expressions) in the time-focused view focuses more on numerical comparisons, i.e., absolute magnitude relationships of the time in time expression.
>    2. Modeling relationships between fact nodes in the fact-focused view focuses more on the sequence of events, i.e., the relative temporal relationship of events.
>    3. We model these two temporal relationships simultaneously through a multi-view mechanism and make absolute and relative temporal relations complementary through a fusion mechanism, which makes time modeling more adequate.
>
> 3. Example:
>    1. Suppose there are two event-time pairs, (go to high school, In 1988) and (go to university, In 1992).
>    2. In the time-focused view, the relation is (In 1998, After, In 1992), which models the absolute relation of time expression (1998 > 1992, so the relation is After).
>    3. In the fact-focused view, the relation is (go to high school, before, go to university), which models the relative temporal relation of events.
>
> **Re: Why to use different baselines in different QA experiments.**
>
> 1. We include BART and DPR in the table simply follows the results of the original paper and has no other implications. If this creates ambiguity and misunderstanding, we can remove these two results from the table.
>
> 2. Our task form is QA over long context, and we need to pick methods that can handle long text input as a baseline. Thus we select FiD and BigBird as the generalized baseline for both datasets. FiD and BigBird are SOTA generative and extractive long text reader model , respectively.
>
> 3. BART and DPR are the baselines for closed-book setting and open-ended setting in the original SituatedQA paper, respectively. Our task form is neither closed-book nor open-ended but QA over long context, therefore FiD and BigBird are the baselines that should be compared.
>
>
>
> **Re: More state-of-the-art QA baselines are needed**
>
> 1. We conduct complementary experiments on two reader models which are capable of handling long text input, LED[1] and LongT5[2].
>
>     The results of the experiment are shown in the table below. Our method still significantly outperforms all baselines.
>
>    [1] Longformer: The Long-Document Transformer
>
>    [2] LongT5: Efficient Text-To-Text Transformer for Long Sequences
>
>   Table1: Experimental results on the TimeQA dataset (* represent complementary experiments).
>
> | Method         | Dev-Easy-EM | Dev-Easy-F1 | Test-Easy-EM | Test-Easy-F1 | Dev-Hard-EM | Dev-Hard-F1 | Test-Hard-EM | Test-Hard-F1 |
> | -------------- | :---------: | ----------- | ------------ | ------------ | ----------- | ----------- | ------------ | ------------ |
> | LED*           |    46.97    | 56.41       | 49.41        | 58.35        | 39.78       | 48.45       | 40.50        | 48.1         |
> | LongT5_newsqa* |    48.17    | 59.59       | 48.14        | 58.94        | 41.43       | 52.58       | 42.94        | 53.50        |
> | LongT5_squad*  |    48.67    | 59.54       | 50.85        | 61.00        | 41.64       | 52.63       | 41.93        | 52.07        |
> | BigBird_nq     |    51.89    | 62.43       | 50.61        | 60.88        | 44.04       | 53.83       | 43.56        | 53.60        |
> | FiD_nq         |    57.25    | 66.56       | 58.06        | 67.52        | 49.21       | 58.00       | 49.80        | 58.74        |
> | **MTGER**      |  **59.65**  | **68.05**   | 60.49        | 69.44        | **52.31**   | 60.66       | 53.19        | 61.42        |
> | **MTGER++**    |    59.20    | 67.70       | **60.95**    | **69.89**    | 52.02       | **61.06**   | **54.15**    | **62.40**    |
>
> Table2: Experimental results on the Situated dataset (* represent complementary experiments). .
>
>
> | **Method**     | **Static** | **Samp.** | **Start** | **Avg.** |
> | -------------- | ---------- | --------- | --------- | -------- |
> | LED*           | 18.1       | 16.2      | 20.4      | 17.8     |
> | LongT5_newsqa* | 22.0       | 14.4      | 18.3      | 16.8     |
> | LongT5_squad*  | 14.5       | 13.4      | 22.0      | 16.2     |
> | BigBird_nq     | 34.2       | 15.7      | 21.3      | 20.2     |
> | FiD_nq         | 28.7       | 22.2      | **31.8**  | 26.4     |
> | **MTGER**    | **45.0**   | **24.8**  | 28.2      | **28.8** |
>
> 2. Explain baseline selection:
>
>    1. Our task is performed on long text, which typically accepts inputs larger than 4000 tokens. This leads to that most QA models would be inapplicable due to the input length limitation. Therefore our chosen baseline methods must be able to handle long text inputs, such as BigBird, LED, LongT5 (by sparse attention mechanism) and FiD (by segmentating document and delaying interactions). Among them FiD and BigBird are generative and extractive SOTA long text readers, respectively.
>
>
>    2. Why don't we select some ODQA methods as a generic baseline, e.g. DPR[3], KG-FiD[4]:
>
>         Open-domain Q&A methods usually use the long-context model as a base reader, and improve the performance of the whole system by retrieval and reranking module.  However, our task is performed over given long documents that do not need retrieving and reranking, so these ODQA methods are equivalent to the long-context reader model when the retrieval and reranking modules are removed. Therefore we do not select methods specifically designed for the ODQA task as general baselines (e.g. KG-FiD[3], which is equivalent to FiD when removing the reranking module and DPR is equivalent to BERT when removing the retrieval module).
>    [3] Dense Passage Retrieval for Open-Domain Question Answering, 2020
>
>    [4] KG-FiD: Infusing knowledge graph in fusion-in-decoder for opendomain question answering, 2022
>
> **Re: The experimental analyses are not sufficient. (Table 5, Results of SituatedQA)**
> 1. I think the reason why our method performs slightly worse on the Start type may be that the model relies too much on the temporal graph to solve the problem.
> 2. Static type questions involve a large number of timestamps, Sampled type questions involve a large number of time intervals. While Start type problems only contain timestamps for the start time of the answer.  In other words, questions of type Start contain fewer time expressions. Since rich temporal graph can be constructed when facing the first two types of question, the model relies on temporal graphs to answer complex questions. However, when faced with Start type questions, the content of temporal graphs is relatively insufficient, resulting in a slight performance degradation.
> 3. Despite the slight performance degradation on Start type, our method significantly outperforms baselines in the remaining types as well as in the overall performance, demonstrating the effectiveness of our method.
>
> **Re: The experimental analyses are not sufficient. (Table 3, Ablation Study)**
>
> We will explain this from both experimental and module design perspectives.
>
>
> |  Method   | Dev-Easy  | Dev-Hard   | Test-Easy  |Test-Hard
> |  ----  | ----  | ----  | ----  | ----  |
> |  MTGER  | **59.65**  | **52.31**  | 60.49  | **53.19**  |
> |  -Time Cmp (original paper) | **59.65**  | 51.53  | **60.66**  | 52.65  |
> |  performance gap | -0.00  | -0.78  | +0.17  | -0.54  |
> |  -Time Cmp (rerun)  | 59.58  | 51.44  | 60.42 | 52.69  |
> |  performance  gap  | -0.07  | -0.87 | -0.07  | -0.5  |
> |  average  gap  | -0.035  | -0.825 | +0.05  | -0.52  |
> 1. Experimental perspective: We rerun the experiment three times and obtained average results as shown above.
>
>    1. As shown above, the effect of removing this module on the performance in the Easy-split is less than 0.1% ($\frac{0.05}{60.49}$), which is negligible.
>    2. Based on the repeated experimental results, we believe that the performance fluctuation comes from random factors, and removing the module does not isignificantly improve or reduce the performance in the Easy-split.
>
>
> 2. Module design perspective
>
>    1. The time comparison module is designed to strengthen the model's implicit reasoning ability, while all of the questions in the easy-split do not need implicit reasoning.
>
>    2. Therefore removing this module should have little or no effect on the performance of the Easy-split. And the experimental results are also in line with our expectations.

---

### Official Review · Reviewer_ZXhY · 2023-08-03

**Soundness:** 3

**Excitement:**

4: Strong: This paper deepens the understanding of some phenomenon or lowers the barriers to an existing research direction.

**Paper Topic And Main Contributions:**

This paper focuses on temporal reasoning over documents, where QA models are required to answer time-related questions based on documents. This paper is inspired by the human reasoning process. Specifically, this paper extracts the temporal relations and discourse relations explicitly and constructs a multi-view heterogeneous temporal graph. Then, it models the relations via a GNN-based graph reasoning component. Finally, it generates the answer via a time-enhanced decoder. The experimental results show the effectiveness of the proposed method.

**Questions For The Authors:**

A) How to obtain the time nodes? Are the time expressions annotated in the datasets, or extracted by another tool?

B) How to judge the relationships between time nodes and fact nodes? Are they annotated in the datasets, or classified by another tool?

C) How to classify the type of time nodes? For example, in Figure 3, what type does the node ``15 years old'' belong to?


**Reasons To Accept:**

1. The idea of constructing the temporal graph sounds reasonable and interesting.
2. This paper proposes a novel framework to fuse temporal and factual information during reasoning.
3. This paper conducts extensive experiments to show the effectiveness of the proposed method.

**Reasons To Reject:**

1. Some details in the graph construction part are not clearly described.
2. The supervised baselines are from 2020. The comparison of more recent baselines, if available, may better demonstrate the effectiveness of the proposed method.


**Reproducibility:**

3: Could reproduce the results with some difficulty. The settings of parameters are underspecified or subjectively determined; the training/evaluation data are not widely available.

**Reviewer Confidence:**

3: Pretty sure, but there's a chance I missed something. Although I have a good feel for this area in general, I did not carefully check the paper's details, e.g., the math, experimental design, or novelty.

---

> ### Author Rebuttal · Authors · 2023-08-27
>
> Thank you for the careful and valuable review, which plays an important role in improving our paper! We address and clarify the questions and concerns below.
>
> **Re: How to obtain the time nodes? Are the time expressions annotated in the datasets, or extracted by another tool?**
>
> 1. We design regular expressions to extract time expressions from documents.
> 2. We treat the extracted TimeX as time nodes, which contain 4 categories  (In, Between, Before, After).  Each time node indicates the time interval in which the event occurred.
> 3. Example(time, time interval): (In 1999, [1999, 1999]; Before March 1988, [~, 1988.25], etc.)
>
> **Re: How to classify the type of time nodes? For example, in Figure 3, what type does the node '15 years old' belong to?**
>
> 1. We categorize time nodes based on the textual features of the temporal expression (e.g. before 2000 is categorized as Before).
>     1. before, until are categorized as Before type.
>     2. between ... and and from ... to are categorized as Between type.
>     3. In, on and at are categorized as In type.
>     4. after and since are categoried as After type.
> 2. For some other types of abstract time expressions, we categorize as In type. (e.g. 1940s, xx years old, xx century, etc.).
>
> **Re: How to judge the relationships between time nodes and fact nodes? Are they annotated in the datasets, or classified by another tool?**
>
> The relations between nodes are rule-based rather than from annotations.
>
> Before introducing how to construct edges, it is necessary to introduce document segmentation.
>
> 1. Document Segmentation: We divide the document into chunks (paragraphs) based on chapter headings, with similar content within each paragraph. (e.g. Early life, College career, Profession career)
> 2. Edge between fact nodes: We construct fact edges according to the Document segmentation. Dense connections (Inter-para Fact Edge) are constructed for fact nodes within the same paragraph (two nodes in the same paragraph), and sparse connections (Intra-para Fact Edge) are constructed between fact nodes across paragraph (two nodes are located in two adjacent paragraphs).
> 3. Edge between time nodes: Each time node can be represented by a time interval (e.g. between 1923 and 1924 can be represented as [1923, 1924]; before 1999 can be represented as [~, 1999]). There are three kinds of interval relations between two time nodes, before, after and overlap. We use this to connect time nodes (Temporal Relation Edge). It is worth noting that the connections between time nodes are also dense within paragraphs and sparse across paragraphs.
>    1. Example (time1, time2, time relation): (In 1999 [1999, 1999], Before 1988 [~, 1988], Overlap), (After 2005 [2005, ~], Between March 2001 and Sep 2002 [2001.25, 2002.75], After)
> 4. Other connections: To prevent graph sparsity, we introduce a global node to aggregate information, and global nodes are connected to all fact nodes (Fact Aggregation Edge). Unidirectional connections are taken from fact nodes to time nodes (Time-to-Fact Edge).
>
> **Re: The supervised baselines are from 2020. The comparison of more recent baselines, if available, may better demonstrate the effectiveness of the proposed method.**
>
> 1. We conduct complementary experiments on two reader models which are capable of handling long text input, LED[1] and LongT5[2].
>
>    The results of the experiment are shown in the table below. Our method still significantly outperforms all baselines.
>
>    [1] Longformer: The Long-Document Transformer, 2020
>
>    [2] LongT5: Efficient Text-To-Text Transformer for Long Sequences, 2022
>
> Table1: Experimental results on the TimeQA dataset (* represent complementary experiments).
>
> | Method         | Dev-Easy-EM | Dev-Easy-F1 | Test-Easy-EM | Test-Easy-F1 | Dev-Hard-EM | Dev-Hard-F1 | Test-Hard-EM | Test-Hard-F1 |
> | -------------- | :---------: | ----------- | ------------ | ------------ | ----------- | ----------- | ------------ | ------------ |
> | LED*           |    46.97    | 56.41       | 49.41        | 58.35       | 39.78       | 48.45       | 40.50        | 48.1         |
> | LongT5_newsqa* |    48.17    | 59.59       | 48.14        | 58.94        | 41.43       | 52.58       | 42.94        | 53.50        |
> | LongT5_squad*  |    48.67    | 59.54       | 50.85        | 61.00        | 41.64       | 52.63       | 41.93        | 52.07        |
> | BigBird_nq     |    51.89    | 62.43       | 50.61        | 60.88        | 44.04       | 53.83       | 43.56        | 53.60        |
> | FiD_nq         |    57.25    | 66.56       | 58.06        | 67.52        | 49.21       | 58.00       | 49.80        | 58.74        |
> | **MTGER**      |  **59.65**  | **68.05**   | 60.49        | 69.44        | **52.31**   | 60.66       | 53.19        | 61.42        |
> | **MTGER++**    |    59.20    | 67.70       | **60.95**    | **69.89**    | 52.02       | **61.06**   | **54.15**    | **62.40**    |
>
> Table2: Experimental results on the Situated dataset (* represent complementary experiments). .
>
>
> | **Method**     | **Static** | **Samp.** | **Start** | **Avg.** |
> | -------------- | ---------- | --------- | --------- | -------- |
> | LED*           | 18.1       | 16.2      | 20.4      | 17.8     |
> | LongT5_newsqa* | 22.0       | 14.4      | 18.3      | 16.8     |
> | LongT5_squad*  | 14.5       | 13.4      | 22.0      | 16.2     |
> | BigBird_nq     | 34.2       | 15.7      | 21.3      | 20.2     |
> | FiD_nq         | 28.7       | 22.2      | **31.8**  | 26.4     |
> | MTGER(ours)    | **45.0**   | **24.8**  | 28.2      | **28.8** |
>
> 2. Explain baseline selection:
>
>    1. Our task is performed on long text, which typically accepts inputs larger than 4000 tokens. This leads to that most QA models would be inapplicable due to the input length limitation. Therefore our selected baseline methods must be able to handle long text inputs, such as BigBird, LED, LongT5 (by sparse attention mechanism) and FiD (by segmentating document and delaying interactions). Among them, FiD and BigBird are generative and extractive SOTA long text readers, respectively.
>
>
>    2. Why don't we select some ODQA methods as a generic baseline, e.g. DPR[3], KG-FiD[4]:
>
>       Open-domain Q&A methods usually use the long-context model as a base reader, and improve the performance of the whole system by retrieval and reranking module.  However, our task is performed over given long documents that do not need retrieving and reranking, so these ODQA methods are equivalent to the long-context reader model when the retrieval and reranking modules are removed. Therefore we do not select methods specifically designed for the ODQA task as general baselines (e.g. KG-FiD[3], which is equivalent to FiD when removing the reranking module and DPR is equivalent to BERT when removing the retrieval module).
>
>        [3] Dense Passage Retrieval for Open-Domain Question Answering, 2020
>
>        [4] KG-FiD: Infusing knowledge graph in fusion-in-decoder for opendomain question answering, 2022

---

### Official Review · Reviewer_dd6k · 2023-08-05

**Soundness:** 3

**Excitement:**

3: Ambivalent: It has merits (e.g., it reports state-of-the-art results, the idea is nice), but there are key weaknesses (e.g., it describes incremental work), and it can significantly benefit from another round of revision. However, I won't object to accepting it if my co-reviewers champion it.

**Paper Topic And Main Contributions:**

This paper presents a multi-view temporal graph enhanced temporal reasoning over time-involved document. The authors also conduct extensive experiments on two time-invoved QA datasets. The research topic is interesting and the solution is clear. The results show that the proposed method outperforms than other baselines.

**Questions For The Authors:**

1. It is not clear how the time-focued graph is constructed. What are the time nodes? According the description on page 3, I guess the edges in the time-focused graph contain 3 different relations? Is it right? If it is right, the graph is very sparse,, which will bring new issue. I suggest the authors give more details and it is neceesary to give one examples to answer the above questions.
2. In Table 1, the authors use gpt-3.5. In the implementation details. the authors use one-single A100 GPU. Can GPT-3.5 run on the single A100?

**Reasons To Accept:**

1. The research topic is interesting and the solution is clear.
2. The results show that the proposed method outperforms than other baselines.

**Reasons To Reject:**

1. It is not clear how the time-focued graph is constructed. What are the time nodes? According the description on page 3, I guess the edges in the time-focused graph contain 3 different relations? Is it right? If it is right, the graph is very sparse,, which will bring new issue. I suggest the authors give more details and it is neceesary to give one examples to answer the above questions.
2. In Table 1, the authors use gpt-3.5. In the implementation details. the authors use one-single A100 GPU. Can GPT-3.5 run on the single A100?

**Reproducibility:**

4: Could mostly reproduce the results, but there may be some variation because of sample variance or minor variations in their interpretation of the protocol or method.

**Reviewer Confidence:**

3: Pretty sure, but there's a chance I missed something. Although I have a good feel for this area in general, I did not carefully check the paper's details, e.g., the math, experimental design, or novelty.

---

> ### Author Rebuttal · Authors · 2023-08-27
>
> Thank you for the careful and valuable review, which plays an important role in improving our paper! We address and clarify the questions and concerns below.
>
> **Re: How the time-focused graph is constructed?**
>
> Constructing the temporal graph is divided into three steps: Document Segmentation, Node Extractions and Edge Construction.
>
>   1. Document Segmentation: We divide the document into chunks(paragraphs) based on chapter headings, with similar content within each paragraph. (e.g. Early life, College career, Profession career)
>
>   2. Node Extraction:
>      1. We design regular expressions to extract Time Expression from documents.
>      2. We treat the extracted TimeX as time nodes, which contain 4 categories  (In, Between, Before, After). The sentence which time nodes are located are treated as fact nodes, corresponding to the time nodes.
>      3. Example: Time node (In 1999, Before March 1988, etc.), fact nodes (He went to UCLA, etc.)
>
>   3. Edge Construction
>      1. Take time-focused graph as example, the graph contains a temporal layer and a factual layer. Figure 3 in the Appendix shows a temporal graph.
>
>      2. Factual Layer: We construct fact edges according to the Document paragraphs divided in the first step. Dense connections (Inter-para Fact Edge) are taken for fact nodes within the same paragraph (two nodes in the same paragraph), and sparse connections (Intra-para Fact Edge) are taken between fact nodes across paragraph (two nodes are located in two adjacent paragraphs).
>
>      3. Temporal Layer: Each time node can be represented by a time interval (e.g. between 1923 and 1924 can be represented as [1923, 1924]; before 1999 can be represented as [~, 1999]). There are three kinds of interval relations between two time nodes, before, after and overlap. We use this to connect time nodes (Temporal Relation Edge). It is worth noting that the connections between time nodes are also dense within paragraphs and sparse across paragraphs.
>
>      4. Other connections: To mitigate graph sparsity problem, we introduce a global node to aggregate information, and global nodes are connected to all fact nodes (Fact Aggr. Edge). Unidirectional connections are taken from fact nodes to time nodes (Time-to-Fact Edge).
>
>
> **Re: What are the time nodes**
>
> 1. The time nodes are time expressions extracted from the document. There are four types of time nodes accordingg to the type of time expression.
>
> 2. Each time node represents a time interval, for example(time, time interval): (In 1999, [1999, 1999]; Before March 1988, [~, 1988.25], etc.)
>
> **Re: Is time-focused graph contain 3 different relations**
>
> The time-focused graph contains a total of 7 types of edge.  Three types in temporal layer (Before, After, Overlap); three types in factual layer (Intra-para, Inter-para, Fact Aggregation); and one type connects the factual and temporal layers (Time-to-Fact).
>
> **Re: Graph sparse problem**
>
> We take two approaches to mitigate the sparsity problem.
>
>   1. We chunk the document by chapter into paragraphs and connect nodes densely within paragraphs and sparsely across paragraphs. This solution alleviates the problem of sparsity while preventing connections from becoming too dense.
>   2. We use global node to aggregate information from fact nodes to alleviate the problem of difficult information interaction between two nodes that are too far apart.
>
> **Re: Question about implementation details**
>
> 1. All supervised experiments (FiD, BigBird, LED, LongT5, MTGER, etc.) were run on a single A100 GPU.
> 2. All LLM experiments (text-davinci-003, gpt3.5) were done by calling OPENAI-API.
>
> **General Response (Complementary experiments)**
>
> We conduct complementary experiments on two reader models which are capable of handling long text input, LED[1] and LongT5[2].
> The results of the experiment are shown in the table below. Our method still significantly outperforms all baselines.
>
>   [1] Longformer: The Long-Document Transformer, 2020
>
>   [2] LongT5: Efficient Text-To-Text Transformer for Long Sequences, 2022
>
> Table1: Experimental results on the TimeQA dataset (* represent complementary experiments).
>
> | Method         | Dev-Easy-EM | Dev-Easy-F1 | Test-Easy-EM | Test-Easy-F1 | Dev-Hard-EM | Dev-Hard-F1 | Test-Hard-EM | Test-Hard-F1 |
> | -------------- | :---------: | ----------- | ------------ | ------------ | ----------- | ----------- | ------------ | ------------ |
> | LED*           |    46.97    | 56.41       | 49.41        | 58.346       | 39.78       | 48.45       | 40.50        | 48.1         |
> | LongT5_newsqa* |    48.17    | 59.59       | 48.14        | 58.94        | 41.43       | 52.58       | 42.94        | 53.50        |
> | LongT5_squad*  |    48.67    | 59.54       | 50.85        | 61.00        | 41.64       | 52.63       | 41.93        | 52.07        |
> | BigBird_nq     |    51.89    | 62.43       | 50.61        | 60.88        | 44.04       | 53.83       | 43.56        | 53.60        |
> | FiD_nq         |    57.25    | 66.56       | 58.06        | 67.52        | 49.21       | 58.00       | 49.80        | 58.74        |
> | **MTGER**      |  **59.65**  | **68.05**   | 60.49        | 69.44        | **52.31**   | 60.66       | 53.19        | 61.42        |
> | **MTGER++**    |    59.20    | 67.70       | **60.95**    | **69.89**    | 52.02       | **61.06**   | **54.15**    | **62.40**    |
>
> Table2: Experimental results on the SituatedQA dataset (* represent complementary experiments). .
>
>
> | **Method**    | **Static** | **Samp.** | **Start** | **Avg.** |
> | ------------- | ---------- | --------- | --------- | -------- |
> | LED*           | 18.1       | 16.2      | 20.4      | 17.8     |
> | LongT5_newsqa* | 22.0       | 14.4      | 18.3      | 16.8     |
> | LongT5_squad*  | 14.5       | 13.4      | 22.0      | 16.2     |
> | BigBird_nq    | 34.2       | 15.7      | 21.3      | 20.2     |
> | FiD_nq        | 28.7       | 22.2      | **31.8**  | 26.4     |
> | **MTGER**   | **45.0**   | **24.8**  | 28.2      | **28.8** |

---

### Meta-Review · Area_Chair_YZoA · 2023-09-19

**Recommendation:** 3

**Metareview:**

Paper Topic And Main Contributions:
* This paper proposes a multi-view temporal graph enhanced reasoning framework for temporal reasoning over time-involved documents, which explicitly models the temporal relationships among facts. The authors conduct extensive experiments on two time-invoved QA datasets.

Reasons to accept:
* The topic is interesting and the solution is clear. The results show that the proposed method outperforms baselines.

Reasons to reject:
* Some details in the graph construction part are not clearly described. The authors explain more details in the rebuttal.
* The supervised baselines are not up-to-date. The authors add results in the rebuttal that show the superiority over additional, newer methods for long-input decoding: LED and LongT5.

---

### Decision · Program_Chairs · 2023-10-07

**Decision:**

Accept-Findings

**Comment:**

Paper Topic And Main Contributions:
* This paper proposes a multi-view temporal graph enhanced reasoning framework for temporal reasoning over time-involved documents, which explicitly models the temporal relationships among facts. The authors conduct extensive experiments on two time-invoved QA datasets.

Reasons to accept:
* The topic is interesting and the solution is clear. The results show that the proposed method outperforms baselines.

Reasons to reject:
* Some details in the graph construction part are not clearly described. The authors explain more details in the rebuttal.
* The supervised baselines are not up-to-date. The authors add results in the rebuttal that show the superiority over additional, newer methods for long-input decoding: LED and LongT5.